# Experimental and Numerical Research on Utilizing Modified Silty Clay and Extruded Polystyrene (XPS) Board as the Subgrade Thermal Insulation Layer in a Seasonally Frozen Region, Northeast China

**Qinglin Li, Haibin Wei, Peilei Zhou \*, Yangpeng Zhang, Leilei Han and Shuanye Han**

School of Transportation, Jilin University, Changchun 130022, China; liql1150142@163.com (Q.L.);
weihb@jlu.edu.cn (H.W.); yangpengz16@mails.jlu.edu.cn (Y.Z.); hanll18@mails.jlu.edu.cn (L.H.);
hansy18@mails.jlu.edu.cn (S.H.)
\* Correspondence: sea_sky_love@yeah.net; Tel.: +86-0431-8509-5370

**Abstract:** For strengthening sustainability of subgrade life-cycle service performance and storing industry solid wastes in seasonally frozen regions, compared to previous research of modified silty clay (MC) which consisted of oil shale ash (OSA), fly ash (FA), and silty clay (SC), we identified for the first time, the variations in the thermal insulation capability of MC with different levels of dry density and moisture content. Taking into consideration the effects of 0–20 freeze-thaw (F-T) cycles by a laboratory test, and by the numerical simulation of coupling moisture-temperature, while considering the effects of F-T cycles, the thermal insulation capability of the MC board and the XPS board were studied quantitatively. The testing results show that the thermal conductivity of MC and SC gradually decreases as the number of F-T cycles increases, and that of the XPS board increases with the increased number of F-T cycles, and tend to be of a constant value of 0.061 W/m/K after 17 F-T cycles. The specific heat capacity of the solid particles of the MC, SC, and XPS board does not change regularly as their moisture content, and the number of F-T cycles change, and their variations are in the range of the test error (2%). Simulation results show that MC has the advantage of the thermal insulation property to reduce the frost-depth of 0.21 m, and the thermal insulation property of the composite layer consisting of the MC and XPS board is greater to reduce the frost-depth of 0.55 m, so that it can protect both the SC and sand gravel of the experimental road from the frost heave damage. The research methods and results are very significant in accurately evaluating the thermal insulation capacity and the sustainability of MC and the composite layer consisting of the MC and XPS board, strengthening the stability of the subgrade and increasing the availability of industrial waste.

**Keywords:** modified silty clay; oil shale ash; fly ash; XPS board; thermal conductivity; specific heat capacity; freeze-thaw cycles; thermal insulation capacity

---

## 1. Introduction

In China, seasonally-frozen soil affects 54% of the land area [1]. The subgrade established in a seasonally frozen region must undergo freeze-thaw (F-T) cycles. In seasonally frozen regions, frost heaving and thawing subsidence, which lead to negative effects on the performance of subgrade across the lifecycle, are common mechanisms of subgrade damage in the F-T cycles [2,3]; further, the subgrade damage causes pavement damage diseases such as longitudinal and transverse cracks, dislocation of cement pavement slabs, breaking in the corner of pavement slabs, and uneven settlement of the pavement slab, and so on [4,5]. Engineering research on seasonally-frozen soil subgrade

concentrates on reducing frost heave and thawing subsidence, e.g., by using non-frost-susceptible base materials, limiting soil moisture content and fine grain content, lowering the groundwater table, and setting the thermal insulation layer [6,7]. Extruded polystyrene (XPS) board is a frequently-used material when engineers install a thermal insulation layer [8]. In Finland, since the 1970s, the XPS board and expanded polystyrene (EPS) board have been used as the frost insulation for tracks in railway structures; however, due to the poor moisture resistance of the EPS board, it was discontinued in 1980 [8]. Cai et al. [9] and Zhao [10] researched the engineering performance of XPS boards and EPS boards on compressive strength, heat insulation, moisture resistance, and their effects on subgrade performance. The results showed that the XPS board was feasible in technology and reasonable in the economy.

Oil shale ash (OSA) is a byproduct of shale oil production [11]. Fly ash (FA), produced during coal combustion, is composed of the particulates and flue gases from coal-fired boilers [12]. The Jinlin province, one important energy consumption and production region, with 27.1743 million residents and Gross Domestic Product (GDP) of 1.5289 trillion yuan in 2015, expended 94.95 million tons of coal and 94,000 tons of shale oil in 2015, and is projected to expend 92.75 million tons of coal and 200,000 tons of shale oil in 2020 [13]. While the exact data for OSA and FA is scarce, based on shale oil and coal consumption, it is projected to be huge. For instance, the Wangqing Oil Shale Industrial Park, Jilin province, generates 105 tons of OSA per year, and the Jilin province possesses another three Oil Shale Industrial Parks of the same scale as that of Wangqing Oil Shale Industrial Park (Figure 1) [14,15].

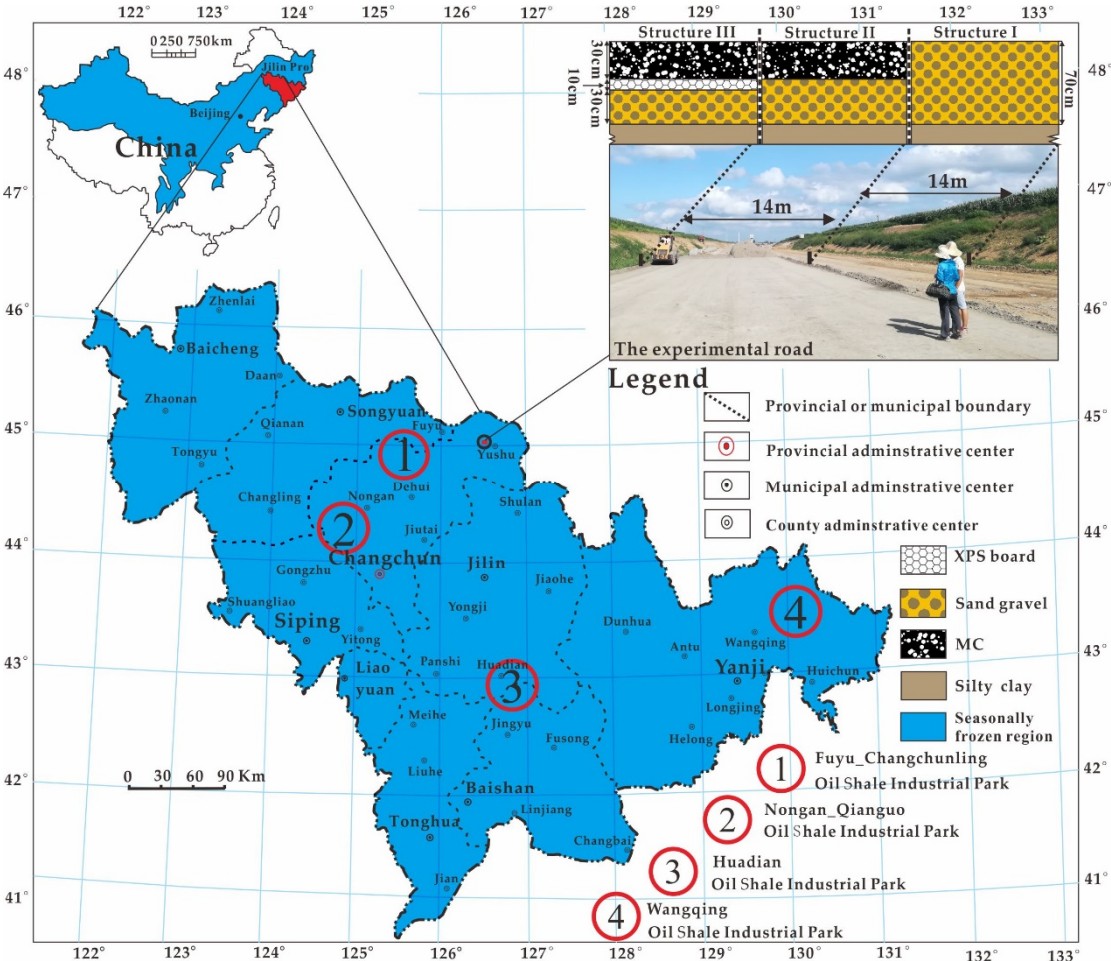

**Figure 1.** The geolocation of the oil shale industrial park and experimental road.

The leachates of OSA and FA landfills are considered to be highly contaminated because they are rich in potentially toxic trace elements, which is congealed from the flue gas, and thus have attracted

much focus and study on the environmental problem [12,16,17]. For reducing the environmental problems by OSA and FA and reusing them, soil modified by them is an effective measure [18,19]. Horpibulsuk and Phetchuay [20] modified silty clay with fly ash and calcium carbide residue, and they classified soil stabilization by calcium carbide residues into three zones: deterioration, inert, active, and research showed that, by adding FA, the strength of the inert zone was increased significantly. Sridharan [21] optimized expansive clay soil using different Classes of FA (ASTM Class C and F) and proportions by weight. The results showed that the swelling potential, compaction properties, and consistency limits of expansive clay soil, improved by FA, were significantly upgraded. It was also observed that 40% of FA content was the best value to optimize the plastic characteristics of expansive clay soil. A few attempts have also been made to stabilize soil with OSA. Mymrin and Ponte [22] undertook an experimental study on the physicochemical interactions for oil-shale fly ash (OSA) with different natural clayey soils. The results indicated that, by changing the percentage of OSA, the strength of the soil could be increased significantly, and modified soil showed very high frost resistance and water resistance as their coefficients equaled or exceeded the 1.0 level. Turner [18] proved that the silty sand, which was processed with OSA, obtained significant improvement in strength, resilient modulus, and F-T durability. In previous literature, research on soil stabilization with both OSA and FA is inadequate, and it focuses on the author's previous research [11,15,23]. Wei et al. [11,23] modified silty clay (SC) by OSA and FA (In the following interpretation, the silty clay, modified by OSA and FA, will be referred to as MC for short), and conducted a battery of laboratory tests to research the mechanical characteristics and environmental impacts of MC; research results showed that the modified SC was suitable for road construction in seasonally frozen areas. Li et al. [15] proved the feasibility of using MC and the XPS board as a road subgrade thermal insulation layer, by the numerical modeling and environmental evaluation on an experiment road located in the Jilin province; however, in their numerical modeling, they viewed the specific heat capacity and thermal conductivity of the MC, XPS board, and SC under F-T cycles as a constant value, so their results could not show the real changes of their thermal insulation performance under freeze-thaw cycles, and thus, their research results exaggerated or belittled the thermal insulation effect of the MC and XPS board. Therefore, research on the effect of F-T cycles on the specific heat capacity and thermal conductivity of the MC, XPS board, and SC is of great significance to accurately evaluate the thermal insulation performance of the MC and XPS board.

Based on the above information, this paper aims to (1) present experimental research on the effect of F-T cycles on the specific heat capacity and thermal conductivity of the MC, XPS board, and SC, (2) show that for MC, XPS board and SC, the improved calculation functions of specific heat capacity and thermal conductivity, considering the effects of F-T cycles, are established, (3) show that, by the numerical method of coupling moisture-temperature calculation, which considers the effect of F-T cycles on the specific heat capacity and thermal conductivity of the MC, XPS board, and SC, the thermal insulation performance, which utilizes the MC and XPS board as the subgrade thermal insulation layer, was identified.

## 2. Materials

### 2.1. Raw Materials

For implementing the experimental research on the effect of F-T cycles on the specific heat capacity and thermal conductivity of the MC, XPS board, and SC, abundant raw materials including OSA, FA, and SC were obtained from the experimental road (Figure 1) and shown in Figure 2, which is located at north latitude 44.896407°, east longitude 126.559129°, and at an altitude of 207.51 m.

In the experimental road, there are two experimental structures named Structure I and II (Figure 1); Structure I is composed of sand gravel and silty clay, and Structure II replaces 30 cm depth of sand gravel by MC, utilized as the subgrade thermal insulation layer. Based on Structures I and II, Structure

III is inspired to present the thermal insulation performance of both MC and XPS board, which is called the novel subgrade thermal insulation layer (NSTIL) in a previous paper by the authors [15].

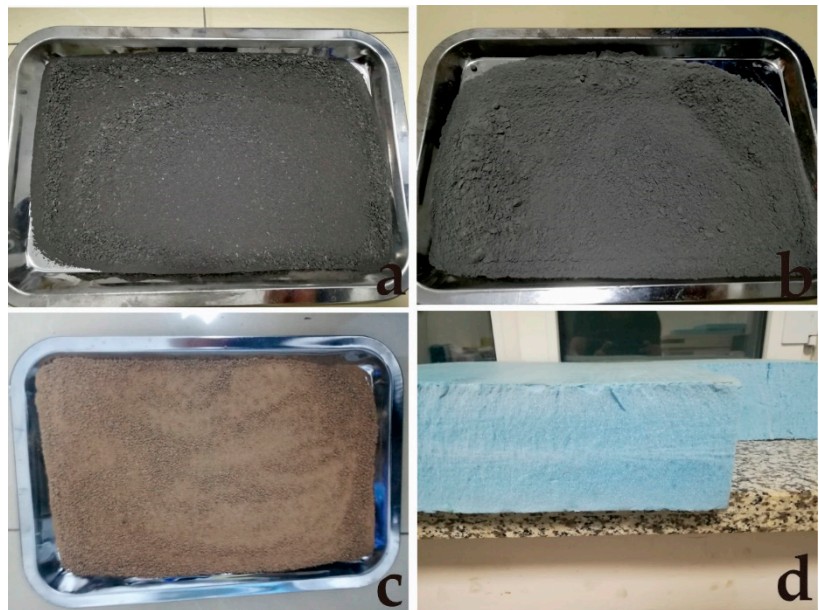

**Figure 2.** The raw materials for OSA (**a**), FA (**b**), SC (**c**) and XPS board (**d**). Notes: (1) OSA = oil shale ash; (2) FA = fly ash; (3) SC = silty clay.

In northeast China, SC is a typical subgrade material. In this study, its plasticity was 13.0%, the plastic limit was 24.3%, and liquid limit was 37.3%. The optimum moisture content and maximum dry density were 12.2% and 1.93 g/cm$^3$, respectively.

For Structures I, II, and III, both information about the source of OSA and FA and their chemical composition, and the source of the XPS board (Figure 2) and its volumetric water absorption, unconfined compressive strength were introduced in Section 2 of reference [15].

### 2.2. Preparation of Mixed Samples

When modifying SC with OSA, FA to manufacture MC, the dry mass ratio of OSA/FA/SC is 2:1:2 and the moisture content is 12.95%; it was determined by two main reasons as follows:

(1) The MC at this ratio possesses great shear strength, dynamic modulus, and the highest CBR value, and better stability after freeze-thaw cycles than unmodified SC, which was certificated by a battery of conventional mechanical and physical property tests [11,23,24]. In this study, its plasticity was 12.40%, liquid limit was 32.50%, and plastic limit was 20.10%; the optimum moisture content and maximum dry density were 12.95% and 1.66 g/cm$^3$, respectively; the CBR value soaking for 96 h was 38%, which was in line with the highway subgrade standards [25].

(2) The strength properties (e.g., CBR value, shear strength, and dynamic modulus) of road subgrade are the most important factors for road operational stability in the highway subgrade standards [25], and the stability of MC after F-T cycles is also worthy of consideration for the life-cycle service performance of subgrade.

Before the experiments, raw materials SC, OSA, and FA, placed in a drying oven, were dried under 105–110 °C for 24 h, and then they were cooled in the desiccator to room temperature. Dry mix soils were mixed in a mass ratio of OSA/FA/SC of 2:1:2. Then, the pure water was added into the dry mix soils and SC with the designed moisture content, which is listed in Table 1, to became wet soils. Sequentially, they were placed in a humidor for three days to allow moisture to seep into the soil evenly.

**Table 1.** The dry density and moisture content of test samples.

| Samples | Moisture (%) | Dry Density (g/cm³) | Number of F-T Cycle (1) | Samples | Moisture (%) | Dry Density (g/cm³) | Number of F-T Cycle (1) |
|---|---|---|---|---|---|---|---|
| MC | 8.00 | 1.59 | 0 | SC | 8.00 | 1.79 | 0 |
| | 10.30 | 1.62 | 0, 3, 5, 10, 20 | | 10.30 | 1.86 | 0, 3, 5, 10, 20 |
| | **12.95** | **1.66** | | | **12.20** | **1.93** | |
| | 15.20 | 1.63 | | | 15.20 | 1.89 | |
| | 18.00 | 1.62 | 0, 20 | | 18.00 | 1.85 | 0, 20 |

Notes: MC = modified silty clay by oil shale ash and fly ash.

## 2.3. Preparation of Test Samples

The XPS board was cut into cubes of 10 cm in diameter as test samples of thermal conductivity and specific surface area. According to the impact molding method [26], soils including MC and SC were made into cylindrical samples (diameter 3.8 cm, height 7.5 cm) with 96% compaction degree, which was also the compaction degree of the experimental road, designed in accordance with the standard of high-grade highway subgrade [25]. The moisture content and dry density of test samples are listed in Table 1, and the bold fonts represent the optimum moisture content and maximum dry density of MC and SC, respectively.

## 3. Methods

### 3.1. Testing Methods

Referring to previous studies [27,28], after seven F-T cycles, the mechanical properties of clay would be stable, so authors thought that, to explore the effect of F-T cycles on the thermal performance (specific heat capacity and thermal conductivity) of the MC, XPS board, and SC, the number of F-T cycles should be greater than or equal to seven. In this experiment, 0, 1,3, 5, 7, 10, 15, and 20 F-T cycles were designed.

F-T cycles of test samples were conducted in Key Laboratories of Road and Bridge, Jilin University, which is also the key laboratory of the Jilin province. The F-T cycles were implemented in a low-temperature test chamber, whose maximum temperature is the indoor temperature, and the minimum is −30 °C with an accuracy of 0.1 °C. For the freezing process, the testing samples were frozen at a temperature ranging from 15 °C to −15 °C quickly, and cooling lasted 24 h. In the thawing process, the temperature was set to 15 °C and a thawing time of 24 h. During F-T cycles, to keep the moisture content of samples unchanged, the samples were wrapped in waterproof films. Visual inspection was carried out before the test to ensure that the surface of the test sample was free from cracks.

The thermal performances of the MC, XPS board and SC were tested at the Jilin University. Thermal conductivity was measured by the Thermal Conductivity Scanner (TCS) with an accuracy of 0.001 W/m/K, which is based on the transient method. Compared with the static test method, the transient method does not need to establish a stable temperature field so that it has two main advantages. Firstly, the static test method would result in soil-water movement in response to temperature gradients, but the transient method does not have that drawback [29]. Secondly, its test time is shorter, and thus, it is suitable for small samples [30]. For the specific heat capacity test, BRR specific heat capacity equipment, with a 2% test error, were employed, which is widely used in the geotechnical thermal properties research with high-precision thermocouple and temperature measuring instruments [33].

### 3.2. Research Methods

In this study, the authors focused on the effect of F-T cycles on the thermal physical performance (thermal conductivity and specific heat capacity) of SC, MC, and XPS board, so the thermal conductivity and specific heat capacity of SC, MC, and XPS board were measured after F-T cycles (0, 1, 3, 5, 7,

10, 15, and 20 times), and based on measured results, the improved calculation function of thermal conductivity and specific heat capacity considering the effects of F-T cycles were established.

In order to highlight the difference between considering the above effect and without that, the numerical simulations of coupling moisture and temperature, considering the effects of freeze-thaw cycles by applying the improved calculation functions of thermal conductivity and specific heat capacity, were conducted for Structures I, II, and III (Figure 1), and the simulation results were compared with the results of Li et al. [15]. The research methods of the improved calculation functions of thermal conductivity and specific heat capacity are explained in Section 3.2.2; the details of the numerical simulations are explained in Section 3.2.3.

### 3.2.1. Numerical Method in the Subgrade during Freeze-Thaw Cycles

In this manuscript, we concentrated on one-dimensional heat transfer and water flow to highlight the difference in thermal insulation performance between considering and not considering the influence of F-T cycles on the thermal physical performances of SC, MC, and XPS board. The mathematical equations of coupling subgrade moisture and temperature during freeze-thaw cycles are composed of Equations (1)–(3):

$$C(\theta)\frac{\partial T}{\partial t} - L \cdot \rho_i \frac{\partial \theta_i}{\partial t} = \lambda(\theta)\nabla^2 T \tag{1}$$

$$\frac{\partial \theta_l}{\partial t} + \frac{\rho_i}{\rho_l} \cdot \frac{\partial \theta_i}{\partial t} = \nabla[D(\theta_l)\nabla\theta_l + k_z(\theta_l)] \tag{2}$$

$$\frac{\theta_i}{\theta_u} = \left\{ \begin{array}{ll} \frac{\rho_w}{\rho_i}\left(\frac{T}{T_f}\right)^B & (T < T_f) \\ 0 & (T \geq T_f) \end{array} \right\} \tag{3}$$

where Equation (1) is the heat transport Equation; (2) is the water flow Equation; (3) is the dynamic balance equation of phase change; the meaning and representation methods of parameters for Equations (1)–(3) have been shown in the authors' earlier paper [15]. In that paper, the volumetric heat capacity of the soil, $C(\theta)$ and the effective thermal conductivity, $\lambda(\theta)$ were denoted by Equations (4) and (5), respectively:

$$C(\theta) = \frac{\theta_s\rho_s C_s + \theta_l\rho_l C_l + \theta_i\rho_i C_i}{\theta_s + \theta_l + \theta_i} \tag{4}$$

$$\lambda(\theta) = (\lambda_s)^{\theta_s}(\lambda_l)^{\theta_l}(\lambda_i)^{\theta_i} \tag{5}$$

where $\theta$, $\rho$, $C$ and $\lambda$ are the volumetric content, density, specific heat capacity, and thermal conductivity, respectively; those subscripts $s$, $l$, $i$ represent solid grains, water, ice, respectively.

In this study, for the sake of considering the effect of F-T cycles on the thermal physical performances of the SC, MC, and XPS board, the Equations (4) and (5) are improved by testing data, whose improved methods will be shown in Section 3.2.2.

### 3.2.2. The improved method of the specific heat capacity and thermal conductivity considering the effects of freeze-thaw cycles

For Equations (4) and (5), in the literature [15], each of $C_s$, $C_l$, $C_i$ and $\lambda_s$, $\lambda_l$, $\lambda_i$ was set to a fixed value, and thus, the variations on $C(\theta)$ and $\lambda(\theta)$ were determined by the changes of $\theta_s$, $\theta_l$ and $\theta_i$. In fact, the specific heat capacity and thermal conductivity of solid, water, and ice were influenced by temperature [31–34]. In this study, the mathematical functions of $C_l$, $C_i$ were summarized from previous literature [29,31–33], listed in Table 2, and $C_s$ was obtained from the analysis of the testing data on the MC, SC and XPS board after F-T cycles. Next, the improved method of $\lambda(\theta)$ is described; in 1987, Chung and Horton [35] proposed the calculation model of soil thermal conductivity:

$$\lambda_0 = a_0 + a_1\theta + a_2\theta^{0.5} \tag{6}$$

where $\lambda_0$ is the soil thermal conductivity; $\theta$ is the soil moisture content; $a_0$, $a_1$, $a_2$ are empirical constants. However, the calculation model of Chung and Horton don't consider F-T cycles as the influencing factor of soil thermal conductivity. For improving the calculation model of Chung and Horton, in this study, the calculation model of the soil thermal conductivity after F-T cycles is established as follows:

$$\lambda = \lambda_0 + \lambda_{F-T} \tag{7}$$

where $\lambda$ is the soil thermal conductivity after F-T cycles; $\lambda_0$ is the soil thermal conductivity of Chung and Horton; $\lambda_{F-T}$ is the correction term of F-T cycles, obtained from the multiple regression methods by Origin 2019 (A trial version).

**Table 2.** The mathematical functions of specific heat capacity summarized from pervious literature.

| | Symbol | Units | Function | $R^2$ | Reference |
|---|---|---|---|---|---|
| Specific heat capacity | $C_l$ (Water) | $J{\cdot}kg^{-1}\,K^{-1}$ | $1020 + 30.637 \times T - 0.0627 \times T^2$ ($274 \leq T \leq 295$ K) | 0.91 | Angell et al. [33] |
| | $C_i$ (Ice) | | $183.24 + 6.9712 \times T$ ($100 \leq T \leq 273$ K) | 0.99 | Dickinson et al. [32] |

Notes: $R^2$ = coefficient of determination.

### 3.2.3. Model Setup

Comsol Multiphysics (version 5.3), a finite element program, was used as the numerical tool. A 64-bit Windows 10 PC, with intel i7 3.1 GHz processor, and 8 GB RAM, was utilized as the computing platform.

For the purpose of emphasizing the difference in thermal insulation performance between considering and not considering the F-T effects on the thermal physical performances of SC, MC, and XPS board, Structures I, II, and III were simulated to get the temperature-moisture distribution, and they were set up as one-dimensional models, whose depths were 10.7 m composed by the replacement thickness of 0.7 m and the thickness of SC for 10 m (Figure 1). The filling materials corresponding to the replacement thickness in Structures I, II, and III, are shown in Figure 1, and the main parameters for the finite-element numerical simulation are listed in Table 3.

**Table 3.** The main parameters of the finite-element numerical simulation.

| | Water Content $w$ [%] | Permeability $k$ [m²] | Density $\rho$ [kg m⁻³] | Specific Heat $C$ [J kg⁻¹ K⁻¹] | Thermal Conductivity $\lambda$ [W m⁻¹ K⁻¹] |
|---|---|---|---|---|---|
| MC | 15.6 | $6.5 \times 10^{-14}$ | 1520 | $C_{MC}$ | $\lambda_{MC}$ |
| XPS board | 0.3 | $1.0 \times 10^{-19}$ | 45 | $C_{XPS}$ | $\lambda_{XPS}$ |
| SC | 23.6 | $5.5 \times 10^{-14}$ | 1640 | $C_{SC}$ | $\lambda_{SC}$ |
| Sand Gravel | 16.0 | $5.5 \times 10^{-12}$ | 1800 | 840 | 1.06 |
| Water | / | / | 980 | $C_l$ in Table 2 | $\lambda_l$ in Table 2 |
| Ice | / | / | 917 | $C_i$ in Table 2 | $\lambda_i$ in Table 2 |

Notes: $C_{MC}$, $C_{XPS}$, $C_{SC}$ and $\lambda_{MC}$, $\lambda_{XPS}$, $\lambda_{SC}$ were obtained from the multiple regression of testing data on MC and SC after F-T cycles, and were shown in Section 4.

For this numerical simulation, boundary conditions and initial values were used. When beginning the simulation, according to the moisture and temperature measured on the experimental road by humidity and temperature sensors, the initial values were set in the models. Information about humidity and temperature sensors, including their setting depth, accuracy, block dimension, and measuring principle, were shown in the literature [15], and the bottom and top boundary conditions of the heat transport and water flow were the same as those of the literature [15], which was the authors' earlier paper.

## 4. Results and Discussion

### 4.1. Variations and Improved Calculation Model on Thermal Conductivity of MC and SC

#### 4.1.1. Variations on thermal conductivity of MC and SC

Figure 3 presents the testing results of the thermal conductivity of MC and SC, and MC-X%-Y and SC-X%-Y both represent X% moisture content and Y g/cm$^3$ density of the MC test sample and SC test sample, respectively. Figure 3A presents variations in the thermal conductivity of MC after F-T cycles; when the number of F-T cycles is zero, the thermal conductivity of MC under different levels of moisture content and dry density possesses the maximum values, and from 0 to 20 F-T cycles, they continue to decrease from 0.938, 0.841, and 0.802 W/m/K to 0.857, 0.799, and 0.768 W/m/K, respectively, indicating that the thermal conductivity of MC decreases with the increased F-T cycles. In Figure 3B, from 0 to 20 F-T cycles, the variation trend of SC thermal conductivity is the same as that of MC; the thermal conductivity of SC with three levels of moisture content and dry density, ranges from 1.570, 1.416 and 1.329 W/m/K to 1.392, 1.350, and 1.307 W/m/K from 0 to 20 F-T cycles, indicating that the thermal conductivity of SC reduces with the increased F-T cycles, too. In Figure 3, it is obvious that, at every level of moisture content and dry density, the thermal conductivity of SC is larger than that of MC. Taking the SC (SC—12.2%–1.93%) and MC (MC—12.95%–1.66%) with the optimum moisture and maximum dry density as an example, corresponding to 0 to 20 F-T cycles, the absolute difference between the thermal conductivity of MC and SC ranges from 0.469 to 0.613 W/m/K, indicating that MC possesses a certain thermal insulation property compared with SC because the smaller thermal conductivity of MC means it takes longer to transfer the same amount of heat.

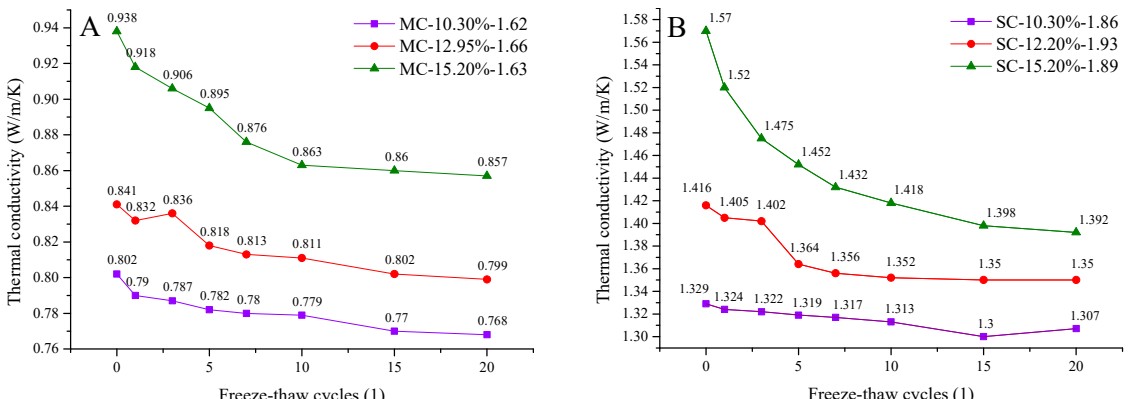

**Figure 3.** The thermal conductivity of MC and SC under F-T cycles. (**A**) The thermal conductivity of MC. (**B**) The thermal conductivity of SC.

#### 4.1.2. Improved Calculation Model on Thermal Conductivity of MC and SC

For improving the calculation model of the thermal conductivity of MC and SC, the thermal conductivity of the MC and SC with five levels of moisture content listed in Table 1 were tested, and then their results were fitted by the multiple regression method by Origin 2019. The fitting process, considering the effect of both moisture content and F-T cycles on thermal conductivity of MC, is shown in Figure 4. Results in Figure 4A show that the calculation model of MC thermal conductivity, without considering F-T cycles, is as follows:

$$\lambda_{MC}(\theta) = 3.052 + 0.256\theta - 1.47\theta^{0.5} \tag{8}$$

where $\lambda_{MC}(\theta)$ is the thermal conductivity of MC considering its moisture content but not F-T cycles; $\theta$ is the moisture content of MC. The coefficient of determination denoted $R^2 = 0.98$ indicates that Equation (8) fits well on the thermal conductivity and moisture content of MC, because in some literatures [36,37], $R^2 \geq 0.90$ is a judgment basis which fit curve is very suitable to represent the

variation trend of fitting data points. Results in Figure 4B show the calculation model of MC thermal conductivity considering F-T cycles as Equation (9):

$$\lambda_{MC} = \lambda_{MC}(\theta) + \lambda_{MC(F-T)} = 3.052 + 0.256\theta - 1.47\theta^{0.5} + 0.98439^{X} + 0.0119X - 1.171 \qquad X = 1,2,3\cdots 20 \qquad (9)$$

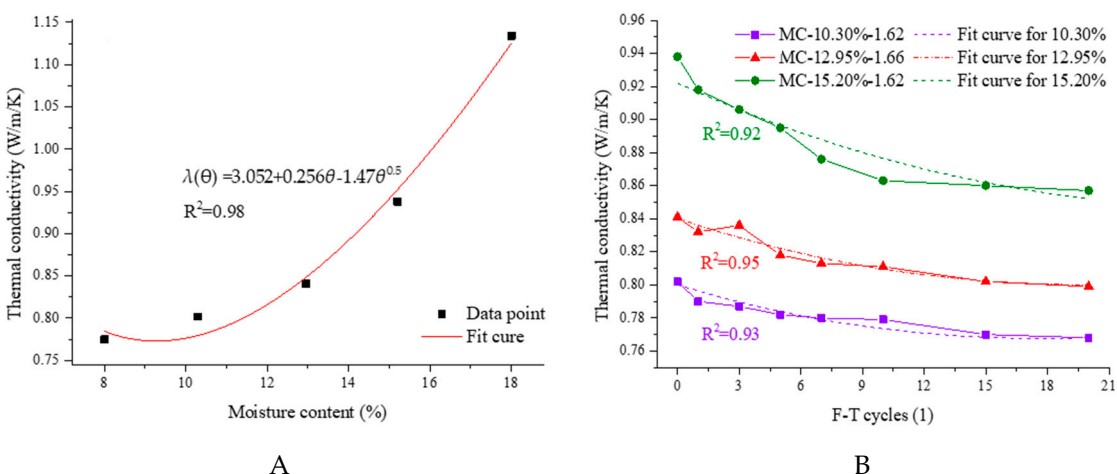

**Figure 4.** Fitting process considering the effect of both moisture content and F-T cycles on thermal conductivity of MC. (**A**) Fitting process of moisture content. (**B**) Fitting process of both moisture content and F-T cycles.

In Equation (9), $\lambda_{F-T} = 0.98439^{X} + 0.0119X - 1.171$ is the mathematical function corresponding to the effect of F-T cycles, and X is the number of F-T cycles ranging from 0 to 20. For samples MC—10.30%–1.62%, MC—12.95%–1.66%, and MC—15.20%–1.62%, from 0 to 20 F-T cycles, the minimum value of $R^2 \geq 0.92$ indicates that Equation (9) fits well on the relation among thermal conductivity, moisture content and F-T cycles of MC. In this manuscript, to implement the numerical simulation of experimental road considering F-T cycles, the time required for each F-T cycle to occur is set as 365 days (one year), so Equations (8) and (9) can be described as Equation (10):

$$\lambda_{MC} = \begin{cases} \lambda_{MC}(\theta) = 3.052 + 0.256\theta - 1.47\theta^{0.5} & t = 0 \quad R^2 = 0.98 \\ \lambda_{MC}(\theta) + \lambda_{MC(F-T)} = \lambda(\theta) + 0.98439^{\frac{t}{365}} + 0.0119\frac{t}{365} - 1.171 & t = 1,2,3\cdots \; m \quad R^2 \geq 0.92 \end{cases} \qquad (10)$$

where *t* is time (day) starting from that medium temperature is lower than 0 °C, and *m* is the maximum of *t*, and it is less than or equal to 7300 (day) because Equations (8)–(10) are obtained from testing data of 20 F-T cycles, and in this manuscript, in order to simulate the F-T cycle condition of the experimental road, each F-T cycle occured for 365 days.

For the thermal conductivity of the SC with five levels of moisture content listed in Table 1, its calculation model is described as Equation (11):

$$\lambda_{SC} = \begin{cases} \lambda_{SC}(\theta) = 2.4899 + 0.1787\theta - 0.9332\theta^{0.5} & t = 0 \quad R^2 = 0.99 \\ \lambda_{SC}(\theta) + \lambda_{SC(F-T)} = \lambda_{SC}(\theta) + 0.95298^{\frac{t}{365}} + 0.0229\frac{t}{365} - 0.9808 & t = 1,2,3\cdots m \quad R^2 \geq 0.91 \end{cases} \qquad (11)$$

In Equation (11), the $R^2$ of $\lambda_{SC}(\theta)$ is equal to 0.99 and that of $\lambda_{SC}(\theta) + \lambda_{SC(F-T)}$ is equal or greater than 0.91, indicating that Equation (11) fits well with the relation among thermal conductivity, moisture content, and F-T cycles of SC.

### 4.2. Variations on Specific Heat Capacity of MC and SC

Figure 5 presents the variations on the specific heat capacity of MC and SC with different levels of moisture content and F-T cycles. It should be noted that the specific heat capacity of MC and SC was

tested after being ground and dried, so testing results showed the specific heat capacity of solid particle in the condition of no contribution of moisture. The specific heat capacity of MC and SC with different levels of moisture content and F-T cycles ranges from 899.87 to 923.99 J/kg/K and 1016.65 to 1042.37 J/kg/K (Figure 5), respectively, and does not change regularly with the changes of moisture content and F-T cycles, and it varies within the range of test error, indicating that the specific heat capacity of solid particle of MC and SC is almost unchanged, and even though it is changed in the condition of different levels of moisture content and F-T cycles, it won't change more than 2%. Based on the above analysis, in the numerical simulation of coupling moisture and temperature, the average values of specific heat capacity of solid particle of MC and SC were used as $C_{MC}$ and $C_{SC}$ listed in Table 3, respectively, and then $C_{MC} = 908$ J/kg/K and $C_{SC} = 1030$ J/kg/K.

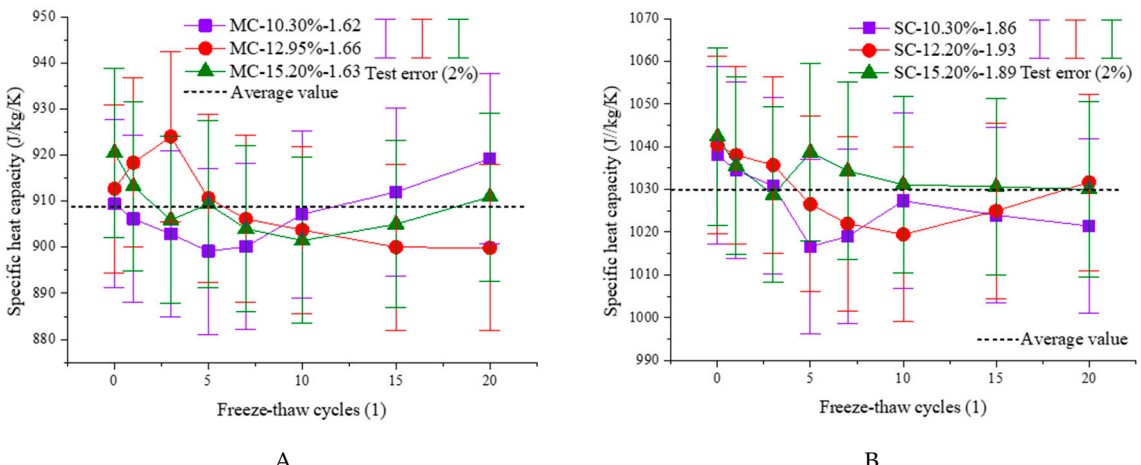

**Figure 5.** Variations on specific heat capacity of MC and SC with different levels of moisture content and F-T cycles. (**A**) The specific heat capacity of MC. (**B**) The specific heat capacity of SC.

*4.3. Variations on Thermal Conductivity and Specific Heat Capacity of the XPS board*

From 0 to 20 F-T cycles, the thermal conductivity of the XPS board increases from 0.030 to 0.061 W/m/K, and its change rates decrease gradually with the increase of F-T cycles, and reaches the maximum value of the 0.061 W/m/K when the number of F-T cycles reaches 17; its calculation model with the function of F-T cycles is great suitable to be described as follows:

$$\lambda_{XPS} = -0.0023\left(\frac{t}{365}\right)^{1.273} + 0.0071\frac{t}{365} + 0.0284 \quad t = 1, 2, 3 \cdots m \quad R^2 = 0.96 \quad (12)$$

where $\lambda_{XPS}$ is the thermal conductivity of XPS board considering the condition of F-T cycles; $R^2 = 0.96$ indicates that Equation (12) represents the change rule of thermal conductivity of XPS board well. The specific heat capacity of the XPS board ranges from 5322 to 5433 J/kg/K from 0 to 20 F-T cycles, and with similar variations on the specific heat capacity of MC and SC in F-T cycles, that of the XPS board also does not change regularly with the changes of F-T cycles (Figure 6B), and it also varies within the range of the test error, so the average value of the specific heat capacity of the XPS board was used as $C_{XPS}$ (5379 J/kg/K) listed in Table 3.

The thermal conductivity of the XPS board is smaller than those of the MC and SC, and its specific heat capacity is larger, indicating that the XPS board possess better thermal insulation capacity because the smaller the thermal conductivity, the more time it takes to transfer the same amount of heat, and the larger specific heat capacity, the more heat it takes to change the same temperature.

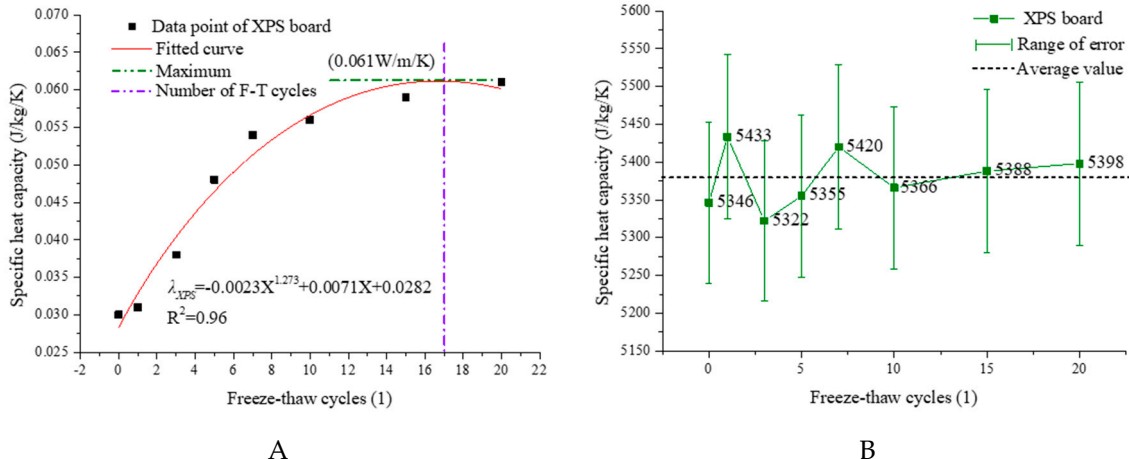

**Figure 6.** Fitting process considering the effect of F-T cycles on the thermal conductivity and specific heat capacity of the XPS board. (**A**) Thermal conductivity. (**B**) Specific heat capacity.

*4.4. Simulation Results Considering the Effects of Freeze-Thaw Cycles on Thermophysical Properties for Structure I*

Figures 7–9 show the simulation results for Structures I, II, and III obtained by the numerical simulation considering the F-T cycles on thermal conductivity and specific heat capacity, which was operated for 1095 days (three years), and data in Figures 7–9 was derived from simulation results from 730 to 1095 days, corresponding dates are 1 July 2020 to 1 July 2021.

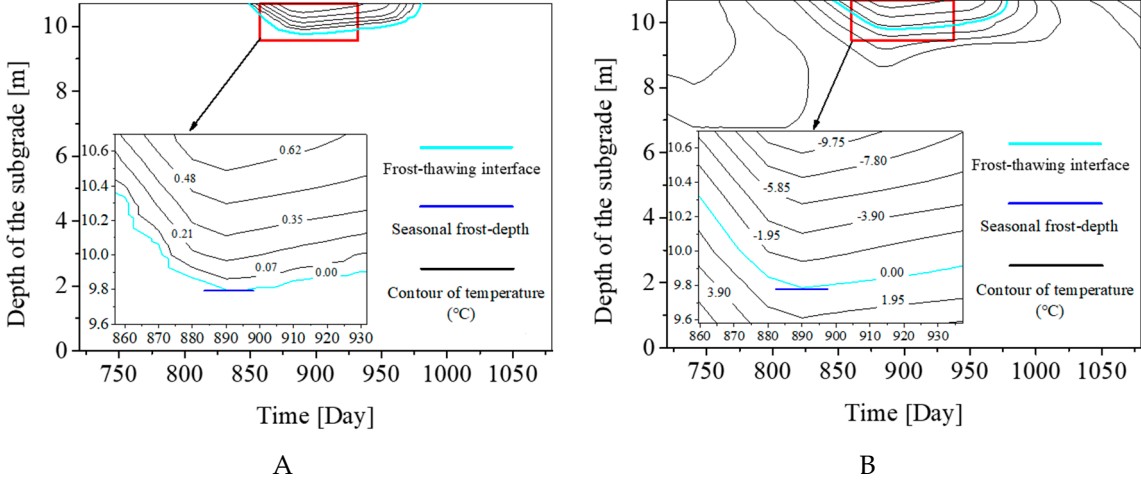

**Figure 7.** The simulation results for Structure I. (**A**) The distribution of ice content. (**B**) The distribution of temperature.

According to Figure 7, the subgrade begins to freeze on the 850th day, and frost-depth reaches the maximum value (1.02 m) on the 890th day, meaning that seasonal frost-depth is 1.02 m for Structure I, and from 890th to 980th day, the subgrade frost-depth ranges from seasonal frost-depth of 1.02 m to 0.0 m. The seasonal frost-depth and frost-thawing interface in Figure 7A is equal to those of Figure 7B, because the temperature and moisture control the ice content of the subgrade. In the freeze-thaw interface domain, the temperature ranges from 0.00 to −11.26 °C (Figure 7B) as the ice content increases from 0.00 to 0.68 (Figure 7A), showing that the ice content increases with the decrease of subgrade temperature below 0 °C.

Compared with simulation results of literature [15], whose frost-depth of Structure I is 1.50 m, the frost-depth in this manuscript decreases to 1.02 m; this is because the numerical model in literature [15] doesn't consider the effect of F-T cycles on thermal conductivity and specific heat

capacity, and thus thermal conductivity and specific heat capacity of the related medium in Structures I, II, and III are set to constants, but the thermal conductivity of medium in Structures I, II, and III is decreasing with the increase of F-T cycles, whose varied trend is shown in Figures 3 and 4; for Structure I, the phenomenon that the thermal conductivity of SC ranges from 2.17 to 2.12 W/m/K (Equation 11) from 0 to 3 F-T cycles is the main reason which leads the frost-depth of Structure I to decrease 0.48 m.

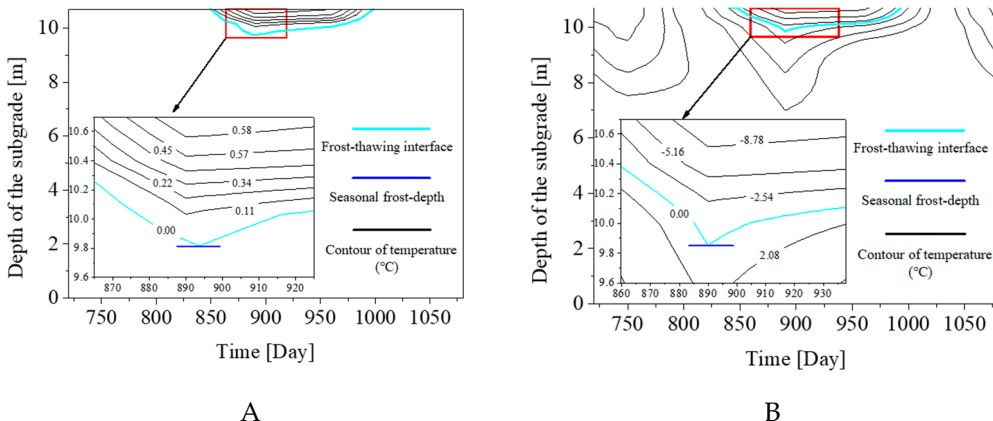

**Figure 8.** The simulation results of Structure II. (**A**) The distribution of ice content. (**B**) The distribution of temperature.

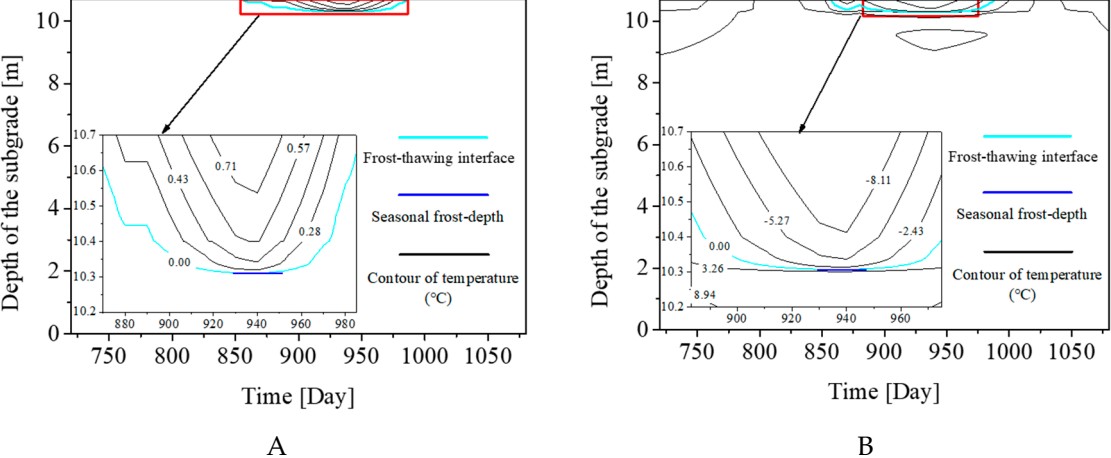

**Figure 9.** The simulation results of Structure III. (**A**) The distribution of ice content. (**B**) The distribution of temperature.

*4.5. Simulation Results Considering the Effects of Freeze-Thaw Cycles on Thermophysical Properties for Structure II*

According to the simulation results of Structure II, in comparison to Structure I, the Structure II, replaced 30 cm depth of sand gravel by the MC (Figure 1), shows the basic thermal insulation capacity.

The Structure II reaches the seasonal frost-depth of 0.81 m on the 890th day (Figure 8), and thus the seasonal frost-depth of Structure II, when compared with that of Structure I, is decreased by 0.21 m. The main reason for the decrease of the seasonal frost-depth on Structure II than Structure I is that the specific heat capacity of the MC is larger and its thermal conductivity is smaller than those of sand gravel (Table 3), and it is well known that, for the same medium, the larger the specific heat capacity it possesses, the more heat it takes to change the same difference in temperature, while the smaller the thermal conductivity is, the longer time is needed to transfer the same amount of heat. The time when Structure II reaches the seasonal frost-depth is the same as that of the Structure I; the main reason for the above is that F-T cycles cause the reduction of thermal conductivity of the SC and MC, which occurs simultaneously.

Compared with the simulation results from the literature [15], whose frost-depth of Structure II is 1.20 m, the frost-depth in this study decreases to 0.81 m; this is because, from 0 to 3 F-T cycles, the thermal conductivity of SC ranges from 2.17 to 2.12 W/m/K (Equation (11)) and the thermal conductivity of MC ranges from 1.57 to 1.475 W/m/K (Equation (10)).

*4.6. Simulation Results Considering the Effects of Freeze-Thaw Cycles on Thermophysical Properties for Structure III*

According to the simulation results of Structure III, in comparison to Structures II and I, the Structure III combines the XPS board and the MC as the subgrade thermal insulation layer, showing the remarkable thermal insulation capacity (Figure 9).

The Structure III also begins to freeze on the 850th day, which is in line with Structures II and I; however, its seasonal frost-depth is less, and the time to reach the seasonal frost-depth is longer. The Structure III gets the seasonal frost-depth of 0.47 m on the 940th day, and thus, the seasonal frost-depth of Structure II, when compared with that of Structures I and II, is decreased by 0.55 m and 0.34 m, respectively, and the time to reach the seasonal frost-depth is 50 days later. The main cause of those differences is that the XPS plate has a smaller thermal conductivity and a higher specific heat capacity than sand gravel and MC (Table 3), and the second is that the permeability coefficient of the XPS plate is close to 0 (Table 3), thus impeding thermal convection.

Compared with the simulation results of literature [15], whose frost-depth of Structure II is 0.60 m, the frost-depth in this study decreases to 0.47 m; this is because, from 0 to 3 F-T cycles, the thermal conductivity of SC ranges from 2.17 to 2.12 W/m/K (Equation (11)), that of MC ranges from 1.57 to 1.475 W/m/K (Equation (10)), and XPS board ranges from 0.03 to 0.40 W/m/K (Equation (12)).

## 5. Conclusions

Based on the above context, the research is novel in several ways based on the following: (1) at different levels of dry density and moisture content, we identified the variations on thermal insulation capability of MC in consideration of the effects of freeze-thaw (F-T) cycles reached 20 by laboratory test for the first time; (2) for the MC, XPS board, and SC, the improved calculation functions of specific heat capacity and thermal conductivity, considering the effects of F-T cycles, are established; (3) by the numerical simulation of coupling moisture-temperature considering the effects of F-T cycles, the thermal insulation capability of MC board and XPS board were studied quantitatively. The specific conclusions are as follows:

According to the test results of thermal conductivity of testing samples with different levels of dry density and moisture content, the MC, which is utilized as the subgrade thermal insulation layer of the experimental road, possesses a certain thermal insulation property compared with SC, and as the number of F-T cycles increases, the thermal conductivity of MC and SC gradually decreases, and the rate of decrease also decreases gradually, but the thermal conductivity of the XPS board is a strong positive correlation with the increase of F-T cycles, and it reaches the maximum value of 0.61 W/m/K when the number of F-T cycles reaches 17.

According to the test results of specific heat capacity of testing samples with different levels of dry density and moisture content, the specific heat capacity of MC, SC, and XPS board doesn't change regularly as their moisture content and number of F-T cycles change, and their variations are in the range of test error (2%), indicating that the specific heat capacity of solid particle of MC, SC, and XPS board is almost unchanged even though the different levels of moisture content and number of F-T cycle are considered, and thus it is reasonable to cite their respective average value for the numerical simulation.

Compared with previous research [15] by the authors of this paper about the thermal insulation capability of MC and XPS board used in Structures I, II, and III, based on the testing data and cited literature, the numerical simulation of the coupling moisture temperature calculation is improved by the fitting equation of thermal conductivity of SC, MC, and XPS board, and by fitting equation of

specific heat capacity of water and ice, and the average value of specific heat capacity for respective solid particle of MC, SC and XPS board. According to the simulation results and compared with previous research [15], the differences are mainly as follows: (1) The seasonal frost depths of Structures III and II are 0.55 m and 0.21 lower than Structure I, respectively, and the time to reach the seasonal frost depths is 50 days later than Structures I and II. (2) Structure III can protect both of SC and sand gravel of the experimental road from the frost heave damage. The reasons for the above difference between this paper and previous research [15] are that with the thermal conductivity of MC, SC, and the XPS board and the specific heat capacity of water and ice, occurs changes with the increased number of F-T cycles, and the numerical model in this paper considers those changes.

The research method and results are of great significance for accurately evaluating the thermal insulation capacity and the sustainability of MC and the composite layer consisting of MC and XPS board, strengthening the stability of subgrade and increasing the availability of industrial waste, and meantime show good sustainability using the XPS board and MC as the subgrade thermal insulation layer in seasonally frozen regions, because the thermal conductivity of MC and SC is negatively correlated with F-T cycles, and despite that of XPS is positively correlated with F-T cycles, it is close to a constant value of 0.061 W/m/K after 17 F-T cycles.

**Author Contributions:** Conceptualization—Q.L.; Data curation—Y.Z.; Funding acquisition—P.Z.; Investigation—Q.L., Y.Z., L.H. and S.H.; Methodology—Q.L.and Y.Z.; Software—L.H. and S.H.; Writing—original draft—Q.L., H.W. and P.Z.; Writing—review & editing—Q.L. and H.W.

**Funding:** This work was supported by the National Key Research and Development Program of China (grant number 2018YFB1600200); National Natural Science Foundation of China (11702108, 51578263).

**Conflicts of Interest:** The authors declare no conflict of interest.

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
