# Peer review of "Experimental and Numerical Research on Utilizing Modified Silty Clay and Extruded Polystyrene (XPS) Board as the Subgrade Thermal Insulation Layer in a Seasonally Frozen Region, Northeast China"

_sustainability, doi:10.3390/su11133495_

Round 1

Reviewer 1 Report

The authors present an interesting and well presented work about the study on thermal performances on utilizing modified silty clay and extruded polystyrene board.

In my opinion the authors present only numerical and calculation models for studing thermal performances of their systems. Though they compare their numerical results with previously paper (Li at al [13]), the authors should better summarize the esxperimental results by adding an appropriate section or paragraph. Also the title of the paper point out on part experimental of the research that de facto has not been descussed.

Author Response

Responses to Reviewers’ Comments on Manuscript ID sustainability-518690

Dear Reviewer 1,

We would like to express our sincere gratitude your thoughtful comments and helpful suggestions for improving the quality of this paper. We have revised the manuscript entitled " Experimental and Numerical Research on Utilizing Modified Silty Clay and Extruded Polystyrene (XPS) Board as the Subgrade Thermal Insulation Layer in a Seasonally Frozen Region, Northeast China " (sustainability-518690) according to your comments. Below are point-by-point responses to your comments. The corresponding modifications and corrections were made and highlighted in red in the revised manuscript (MS).

If there is any question regarding this version of the manuscript, please let us know. We are looking forward to receiving your evaluation.

Please you download the PDF

Best regards,

Peilei Zhou

Jilin University

Changchun, China

Comment 1:

In my opinion the authors present only numerical and calculation models for studying thermal performances of their systems. Though they compare their numerical results with previously paper (Li at al [13]), the authors should better summarize the experimental results by adding an appropriate section or paragraph.

Response:

After thinking over your suggestion from Comment 1, we have revised this manuscript to further summarize the experimental results from lines 361-362 and 439-442 by a few descriptions. We havent made a lot of changes to this manuscript for the following reasons:

 (1) about experimental content of this manuscript, at different levels of moisture content and dry density, we conducted the research about the thermal conductivity and specific heat capacity of MC (silty clay modified by oil shale ash and fly ash) and SC (silty clay) by methods of laboratory test, and the related research results were shown in sections 4.1 and 4.2; meantime, we also researched the thermal conductivity and specific heat capacity of XPS board by means of experimentation and related research results were shown in section 4.3.

(2) in fact, authors think that sections 4.1 to 4.3 explain experimental results suitably, and adding the other section or paragraph to them could be reviewed as be redundantly.

(3) finally, we consider the reason which dear reviewer think that we lack of analysis for experimental results could be because you view the analysis about experimental data from sections 4.1 to 4.3 as numerical and calculation models, but it's also actual an analysis of the test data.

If you insist what we should do, please tell us and we will do.

Comment 2:

Also the title of the paper point out on part experimental of the research that de facto has not been discussed.

Response:

Thanks very much for your suggestions. However, we didnt revise the title of the manuscript, because the research of this manuscript is composed of numerical calculation and experimental analysis, as explained in the response of Comment 1, the experimental analysis from laboratory test is the main component.

We think, if we revise the title Experimental and Numerical Research on Utilizing Modified Silty Clay and Extruded Polystyrene (XPS) Board as the Subgrade Thermal Insulation Layer in a Seasonally Frozen Region, Northeast China to indicating that it refers to a study of a numerical research, that it wont show the function of experimental research, and this kind of topic may deviates from the original intention of our study.

So, please you forgive us not to revise the title of manuscript by your suggestions, but if you insist what we should do, please tell us and we will do.

Reviewer 2 Report

Please highlight the novelty of the research. 

Provide some images for materials used specially for XPS.

Add references for all standard such as ASTM etc.

Provide summarized information about the effects of freeze-thaw (F-T) cycles on subgrade and pavement performance.

Author Response

Responses to Reviewers’ Comments on Manuscript ID sustainability-518690

Dear Reviewer 2,

We would like to express our sincere gratitude your thoughtful comments and helpful suggestions for improving the quality of this paper. We have revised the manuscript entitled " Experimental and Numerical Research on Utilizing Modified Silty Clay and Extruded Polystyrene (XPS) Board as the Subgrade Thermal Insulation Layer in a Seasonally Frozen Region, Northeast China " (sustainability-518690) according to your comments. Below are point-by-point responses to your comments. The corresponding modifications and corrections were made and highlighted in red in the revised manuscript (MS).

If there is any question regarding this version of the manuscript, please let us know. We are looking forward to receiving your evaluation.

Please you download the PDF

Best regards,

Peilei Zhou

Jilin University

Changchun, China

Comment 1:

Please highlight the novelty of the research.

Response:

According to your suggestions, we have revised this manuscript and highlighted the novelty of this research in Conclusion from lines 519 to 526.

Comment 2:

Provide some images for materials used specially for XPS.

Response:

At line 178, we have added related images for materials used (Figure 2).

Comment 3:

Add references for all standard such as ASTM etc.

Response:

We have added references for all standard, and it should be noted that the above all references are Chinese technology standard, and the reason which we apply Chinese technology standard to make related discussion is that our research object come from the seasonally frozen region, northeast China.

Comment 4:

Provide summarized information about the effects of freeze-thaw (F-T) cycles on subgrade and pavement performance.

Response:

We added related content from lines 36 to 39.

Reviewer 3 Report

Dear authors,

Please clearly write (abstract and conclusions) that the study relate to a maximum of 20 cycles of the F-T. In line 186 formula (1) is editorial error. Are the vertical scale on Figures 6,7,8 should not be reversed ? Cylindrical samples (diameter 3.8 cm, height 7.5 cm) are small. What about the effect of scale. Have you checked the effect of sample size on results ?

Author Response

Responses to Reviewers’ Comments on Manuscript ID sustainability-518690

Dear Reviewer 3,

We would like to express our sincere gratitude your thoughtful comments and helpful suggestions for improving the quality of this paper. We have revised the manuscript entitled " Experimental and Numerical Research on Utilizing Modified Silty Clay and Extruded Polystyrene (XPS) Board as the Subgrade Thermal Insulation Layer in a Seasonally Frozen Region, Northeast China " (sustainability-518690) according to your comments. Below are point-by-point responses to your comments. The corresponding modifications and corrections were made and highlighted in red in the revised manuscript (MS).

If there is any question regarding this version of the manuscript, please let us know. We are looking forward to receiving your evaluation.

Please you download the PDF

Best regards,

Peilei Zhou

Jilin University

Changchun, China

Comment 1:

Please clearly write (abstract and conclusions) that the study related to a maximum of 20 cycles of the F-T.

Response:

In lines 15 and 521, we have revised this manuscript according to yours.

Comment 2:

In line 186 formula (1) is editorial error.

Response:

Thanks for your careful observation, and we have revised that fault in line 263

Comment 3:

Whether the vertical scale on Figures 6,7,8 should not be reversed?

Response:

Firstly, in the revised manuscript, the Figures 6, 7 and 8 have become Figures 7, 8 and 9. Secondly, the vertical scale on Figures 7, 8 and 9 is right, and the data, used to draw Figures 7, 8 and 9, was obtained from numerical model which set vertical scale as that of Figures 7, 8 and 9. Finally, authors think the method of expression on vertical scale of Figures 7, 8 and 9 is easy to understand, so please forgive us not to revise this manuscript by your Comment 3, but if you insist what we should do, please tell us and we will do.

Comment 4:

Cylindrical samples (diameter 3.8 cm, height 7.5 cm) are small. What about the effect of scale. Have you checked the effect of sample size on results?

Response:

Authors think the effect from small samples on the results of thermal conductivity is low. The main reasons are as follows:

(1) It is because that the samples scale is small, but it is satisfied with the test requirement of Thermal Conductivity Scanner (TCS) which is based on the transient method because that method is suitable for small samples, and more details can be found in lines 227 to 233.

(2) In fact, in the initial laboratory phase, we measured the thermal conductivity of 3 big samples (diameter 10 cm, height 10 cm) and took the comparison of thermal conductivity between dig samples (diameter 10 cm, height 10 cm) and 3 small samples (diameter 3.8 cm, height 7.5 cm); results shown that the all difference values are less than 0.005 W/m/K.

For specific heat capacity, the test samples are powders, and the scale effect can be ignored.

Figure 1. Big samples (diameter 10 cm, height 10 cm).

This manuscript is a resubmission of an earlier submission. The following is a list of the peer review reports and author responses from that submission.